# Impact of SO$_2$ Flux Estimation in the Modeling of the Plume of Mount Etna Christmas 2018 Eruption and Comparison against Multiple Satellite Sensors

Claire Lamotte [1], Virginie Marécal [1,*], Jonathan Guth [1], Giuseppe Salerno [2], Stefano Corradini [3], Nicolas Theys [4], Simon Warnach [5], Lorenzo Guerrieri [3], Hugues Brenot [4], Thomas Wagner [5] and Mickaël Bacles [1]

1 Centre National de Recherches Météorologiques, CNRM, Université de Toulouse, Météo-France, CNRS, 31057 Toulouse, France
2 Istituto Nazionale di Geofisica e Vulcanologia, INGV, Osservatorio Etneo, 95125 Catania, Italy
3 Istituto Nazionale di Geofisica e Vulcanologia, INGV, ONT, 00143 Roma, Italy
4 Royal Belgian Institute for Space Aeronomy, BIRA-IASB, 1180 Brussels, Belgium
5 Max Planck Institute for Chemistry, MPIC, 55128 Mainz, Germany
* Correspondence: virginie.marecal@meteo.fr

**Abstract:** In this study, we focus on the eruption of Mount Etna on Christmas 2018, which emitted great amounts of SO$_2$ from 24th to 30th December into the free troposphere. Simulations based on two different estimations of SO$_2$ emission fluxes are conducted with the chemistry-transport model MOCAGE in order to study the impact of these estimations on the volcanic plume modeling. The two flux emissions used are retrieved (1) from the ground-based network FLAME, located on the flank of the volcano, and (2) from the spaceborne instrument SEVIRI onboard the geostationary satellite MSG. Multiple spaceborne observations, in the infrared and ultraviolet bands, are used to evaluate the model results. Overall, the model results match well with the plume location over the period of the eruption showing the good transport of the volcanic plume by the model, which is linked to the use of a realistic estimation of the altitude of injection of the emissions. However, there are some discrepancies in the plume concentrations of SO$_2$ between the two simulations, which are due to the differences between the two emission flux estimations used that are large on some of the days. These differences are linked to uncertainties in the retrieval methods and observations used to derive SO$_2$ volcanic fluxes. We find that the uncertainties in the satellite-retrieved column of SO$_2$ used for the evaluation of the simulations, linked to the instrument sensitivity and/or the retrieval algorithm, are sometimes nearly as large as the differences between the two simulations. This shows a limitation of the use of satellite retrievals of SO$_2$ concentrations to quantitatively validate modeled volcanic plumes. In the paper, we also discuss approaches to improve the simulation of SO$_2$ concentrations in volcanic plumes through model improvements and also via more advanced methods to more effectively use satellite-derived products.

**Keywords:** SO$_2$ flux estimation; Mount Etna eruption; modeling; volcanic plume dispersion; multi-sensor comparison



## 1. Introduction

Volcanic emissions, and especially volcanic eruptions, are of serious interest due to their impact on atmospheric chemistry at global and regional scales [1] and sometimes on air quality at a local scale [e.g., [2–7]]. While water vapor (H$_2$O) and carbon dioxide (CO$_2$) are the main gases emitted by volcanoes, their detection remains challenging because they are present in great amounts in the atmosphere. Sulfur dioxide (SO$_2$), also emitted in large quantities by volcanoes, is easier to detect and represents a good proxy for the volcanic

activity. Moreover, sulfur dioxide leads to the formation of sulfate aerosols, which are compounds affecting atmospheric composition and the climate.

Therefore, vulcanologists have developed tools to observe and measure volcanic $SO_2$, which include a recent focus on spaceborne instruments. Indeed, instruments onboard satellites can detect emissions and plumes from volcanoes all around the world, including hard-to-access volcanoes. Those instruments can measure the $SO_2$ amount in the atmosphere in several spectral bands, such as infrared (IR), ultraviolet (UV) and microwave. The retrieved quantity from those instruments is generally the column concentrations of volcanic $SO_2$. Then, from these data, the estimation of the emission flux can be deduced by applying different methods [e.g., [7–12]].

The estimation of the volcanic $SO_2$ column concentrations depends on the instrument and the technique used for the retrieval that can lead to differences. For example, discrepancies can exist by using an IR or a UV instrument. IR instruments present the advantage to detect $SO_2$ on the whole globe during both daytime and nighttime. However, the absorption lines of $SO_2$ overlap with those of water vapor, which strongly absorbs in the IR. In general, at low altitude, where the content in water vapor is more important, approximately below 5 km, sulfur dioxide is less detectable [13]. On the contrary, UV instruments can detect $SO_2$ across the whole troposphere. However, the detection is limited by the presence of sunlight. Therefore, it is impossible to obtain observations of $SO_2$ at night, leading, for instance, to a lack of data over high latitudes in winter.

Besides the observations, models can be useful tools to study volcanic plumes and to establish the link between the flux estimations and the amount measured in the volcanic plume. By knowing, for a specific eruption, the temporal flux estimation and plume height, models can simulate the spatio-temporal evolution of a volcanic plume at the regional or global scale for the study of its transport and its chemical composition. A satellite-derived $SO_2$ column can be used to evaluate the model performances regarding the location of volcanic plumes and their concentrations [e.g., [14]].

In this paper, we study the specific eruption of Mount Etna around Christmas 2018. This was a well-documented case study for which several $SO_2$ emission flux estimations are available [11,12]. We analyze the impact on the modeling of the volcanic plume by using two of these flux estimations. We evaluate the modeled volcanic plume by comparing the results with $SO_2$ column estimations from multiple sensors (IASI, OMI, and TROPOMI). This serves as the basis to discuss the limitations of the parameters retrieved from the observations (emission flux and height, plume concentration, etc.) and their impact on the modeling of volcanic plumes [15,16].

In the following section of this paper, the chosen case study is presented. Then, in Section 3, a description of all the observations used for this work is provided; this includes both the observations used to set the volcanic emissions implemented in the model and the observations used to evaluate the simulations. The MOCAGE model and the simulations conducted are described in Section 4. The model results are presented and evaluated in Section 5, before the discussion and conclusion are presented in Sections 6 and 7, respectively.

## 2. Case Study: The Mount Etna Christmas 2018 Eruption

Mount Etna is a strato-volcano with multiple craters based on the east coast in Sicily (Italy, 37°45′18″N, 14°59′42″E). It is among the most active volcanoes in the world. Mount Etna is an important source of $SO_2$ into the atmosphere, mostly by passive degassing but this is sometimes interrupted by eruptive phases [17]. Its height is 3330 m a.s.l.

The eruption of Christmas 2018 was preceded by mild eruptive activity with small lava flows [18–21]. Then, in the morning of 24th December, the eruptive activity strengthened and a large breach occurred on the south-east flank of the volcano. At 11:15 UTC, a large column of ash and $SO_2$ was emitted into the troposphere at a maximum altitude of 8 km. The strong eruptive activity lasted until 24th December afternoon, but the eruption continued until 30th December with the top of the eruptive column varying from 4 to

5.5 km altitude (complete description of the eruption in [11]). Therefore, this eruption, which injected $SO_2$ in the free troposphere, had no impact on air pollution at the surface even at a local scale over Sicily. Note that, with its maximum reaching 8 km altitude, this is not an exceptionally large plume height for Mount Etna [22].

This volcanic eruption is chosen for several reasons. Firstly, a large amount of $SO_2$ was emitted directly into the troposphere for a few days, but only a small quantity of ash was (with exception of 24th December). Therefore, the detection of $SO_2$ from spaceborne instruments is easier and more accurate. Secondly, due to very variable wind conditions, the temporal evolution of the volcanic plume was complex (change of direction and vertical wind shear). Thirdly, during the eruption, a large number of instruments measuring $SO_2$ in different spectral bands were available. Finally, this eruption is a well-documented case, which was studied in detail in [11,12]. In these studies, the parameters of the eruption (plume height, ash, and $SO_2$ flux) were estimated using the SEVIRI instrument and compared against the estimations made from the FLAME ground-based data (see Section 3). Even though there is an overall good agreement of the two instruments, some differences remain between the retrieved fluxes [11]. The SEVIRI's $SO_2$ flux estimate was also compared with estimates from other spaceborne instruments (Aqua/Terra-MODIS, NPP/NOAA20-VIIRS, Sentinel5P-TROPOMI, MetopA/MetopB-IASI, and Aqua-AIRS) and shows a generally good agreement [12]. However, some differences are visible, caused by the measurement sensitivity of each instrument and the input parameters (plume height or wind speed) used by the flux retrieval algorithm.

## 3. Observations

A large number of observational data are available to study this eruption. In the two following sections, we present those used in this work. On the one hand, we use the estimations of the volcanic emission flux from two very different types of observations (FLAME and SEVIRI) and, on the other hand, we use several observational data to evaluate the simulated $SO_2$ concentrations in the plume during its transport.

### 3.1. $SO_2$ Flux Estimation of the Eruption

Firstly, we present the data that were used to define the emission source of the eruption ($SO_2$ flux and plume height) in the model:

- The ground-based network FLAME (FLux Automatic Measurement) is composed of 9 DOAS spectrometers measuring $SO_2$ in ultraviolet bands (UV) [23]. This network is managed at the observatory of Mount Etna (INGV). The nine spectrometers are located all around the volcano, on its flank (altitude around 900 m a.s.l.), and are spaced from one another by about 7 km (Figure 1 from [11]). Each instrument scans the sky for about 9 h and crosses the volcanic plume at a distance of 14 km from the craters. The complete scan with all the instruments provides a UV spectrum each 5 min, almost in real-time. Then, the transmitted spectra are analyzed using the DOAS technique with a clear sky standard spectrum. From these data, $SO_2$ emissions fluxes are calculated. The uncertainty associated with those data are estimated between $-22\%$ and $+36\%$ [11]. The estimates of $SO_2$ emission flux from FLAME are available a few hours per day with a frequency from 5 to 15 min from 24th December at 8:10 UTC to 30th December at 12:41 UTC.

- SEVIRI (Spinning Enhanced Visible and Infrared Imager) is a spaceborne instrument onboard the geostationary satellite MSG (Meteosat Second Generation). It measures $SO_2$ in infrared bands and has a spatial resolution of 3 km × 3 km at nadir. The instrument has two temporal resolutions depending on the scanning mode: 5 min in a small area over Europe and North Africa (rapid scan) and 15 min for the entire hemisphere (full disk). $SO_2$ emission flux is calculated using the wind speed simulated by the hydro-meteorological model of ARPA (Agenzia Regional per la Protezione Ambientale) interpolated at the plume height and the $SO_2$ quantity retrieved at each pixel of SEVIRI using the VPR (Volcanic Plume Retrieval) procedure (more details

in [11,12]). The uncertainty associated with those data is estimated at 45% [12]. Note that the effect of the ash is to absorb overall the TIR spectral range, then also in the channels used for the $SO_2$ retrievals (8.7 microns). Even if the algorithm is designed to correct for this effect, an overestimation of the $SO_2$ retrieved is still possible. The estimation of $SO_2$ emission flux from SEVIRI is available from 24th December at 10:49 UTC to 30th December at 14:57 UTC each 15 min, except on the 25th. It has been validated by using many different observations collected from several polar satellite sensors such as MODIS, VIIRS, TROPOMI, IASI, and AIRS [12]. The plume height estimation is obtained by using the "dark pixel" method [24]. This method is based on the comparison between the minimal brightness temperature at 10.8 μm of a fixed pixel located over Mount Etna's summit crater and the vertical profile of temperature measured in the same area and at the same time. Due to the relatively small thickness of the volcanic plume, the "dark pixel" method could only be used on 24th December when the plume top was the highest.

- From 25th to 30th December, the volcanic plume height was retrieved using a ground-based network of visible cameras. There are two stations: one in Catania on the south flank of the volcano and another in Bronte on the west flank. By knowing the wind speed and direction, it is possible to retrieve the plume height on the camera's recorded footage [11]. The uncertainty associated with those data is estimated at ±500 m [25].

The two datasets of flux emissions are chosen because they are based on different types of instruments and methodologies. The first one, the spaceborne instrument SEVIRI, provides data at a high frequency over almost the whole eruptive period. The second one, the ground-based network FLAME, provides estimations of the flux emission from measurements very close to the volcanic vent, with a similar frequency as SEVIRI, but is not available at all at night. Moreover, FLAME emission flux is the most different estimate compared to SEVIRI for this case study [11]. This way, we can analyze the maximum uncertainty on the emissions related to the choice of instrument. Additionally, note that information on the uncertainty values of the flux estimates is available for both instruments.

In this section, we have provided the main information on FLAME and SEVIRI products. A comprehensive description and discussion of the procedures adopted for the $SO_2$ flux retrievals and their uncertainties have been described in [11,12].

### 3.2. $SO_2$ Plume Concentrations

The retrievals of the $SO_2$ column from spaceborne observations were compared against the model simulations to evaluate the representation of the plume's transport as well as the estimation of $SO_2$ concentrations within the plume. We used the following products:

- The $SO_2$ concentration columns retrieved by the instrument TROPOMI (TROPOspheric Monitoring Instrument) onboard the Sentinel-5 Precursor [26] are available since 2018. The spatial resolution of the instrument is $3.5 \times 7.2$ km$^2$ for this study. After the first measurement period, during 2019, its spatial resolution was improved to $3.5 \times 5.5$ km$^2$ at nadir. Its temporal resolution over the Mediterranean region is of one or two overflies per day (around from 11 to 12 UTC). Sulfur dioxide is measured by TROPOMI in the UV band. In this work, we use two datasets. The first one, named TROPOMI_OP, corresponds to the operational product [27], which uses a retrieval algorithm based on the DOAS method (Differential Optical Absorption Spectroscopy). The uncertainty associated with those data is estimated at 35% [26]. Here, we use the $SO_2$ column interpolated at 5 km altitude from the 1 km and 7 km products. The 5 km altitude is chosen because it corresponds of the mean altitude over the whole eruption period. The second one, named TROPOMI_MPIC, is a personal communication from the Max Planck Institute for Chemistry (MPIC), which corresponds to a similar algorithm as the operational one but is based on [28] and was used as the verification algorithm for TROPOMI [27]. As for TROPOMI_OP, the $SO_2$ column retrieval assumes a plume altitude of 5 km. The uncertainty associated with the TROPOMI_MPIC product is also estimated at 35%. Both TROPOMI_OP and TROPOMI_MPIC algorithms

are optimized for the analysis of strong and variable volcanic plumes. In particular, they use a combination of different fit windows depending on the strength of the $SO_2$ absorption. However, the exact choices of the wavelength ranges and the transition thresholds are different. Depending on the specific properties of the Etna plume (e.g., the $SO_2$ column and the plume altitude), one of the two algorithms might be better suited, and the inclusion of both algorithms in the comparison better covers the possible range of retrieval results.The small differences in the analysis settings between TROPOMI_OP and TROPOMI_MPIC are detailed in Appendix A.

- The total $SO_2$ columns retrieved using the OMI (Ozone Monitoring Instrument) instrument onboard the Aura [29,30] satellite have been available since 2004. Their spatial resolution is $13 \times 24$ km$^2$ at nadir and their temporal resolution over the Mediterranean region is of one or two overflies per day (around from 11 to 12 UTC). Similar to TROPOMI, sulfur dioxide is measured in the UV band. The uncertainty associated with those data is estimated at 30% [31]. The retrieval algorithm is based on a different method than the one used for TROPOMI's products; the PCA method is used instead (Principal Component Analysis) [29].

- The total columns of $SO_2$ retrieved using the IASI instrument (Infrared Atmospheric Sounding Interferometer) onboard Metop-A and Metop-B [13,32] have been available since 2006 and 2012, respectively. The spatial resolution of the instrument is a circle of 12 km diameter at nadir and its temporal resolution over the Mediterranean region is about four overflies a day (two around from 08 to 09 UTC and two around from 19 to 21 UTC). Unlike TROPOMI and OMI, sulfur dioxide is measured in the IR band, which means that it is sensible up to the pole. However, the sensitivity below 5 km altitude is strongly reduced. The uncertainty associated with those data is estimated at 50% [13].

## 4. Model and Simulation Description

### 4.1. MOCAGE Chemistry-Transport Model

MOCAGE (Modèle de Chimie Atmosphérique à Grande Échelle) is the offline global and regional three-dimensional chemistry-transport model (CTM) developed at CNRM [33,34]. Its applications are very diverse: effect of climate change on atmospheric composition [e.g., [35–38]], global impact of biomass burning [39] and of volcanic emissions [1] on air composition, or operational use for the forecasting of air quality both for France (Prev'Air program [40]) and for Europe [41].

As an offline model, the meteorological fields (wind speed and direction, temperature, humidity, pressure, precipitation, and cloud) used as inputs in MOCAGE are from an independent numerical weather prediction model. MOCAGE can produce simulations with global and regional resolutions thanks to its grid nesting capability (from $2° \times 2°$ to $0.1° \times 0.1°$). The outer domain forces the inner domain at its edges (boundary conditions). Its vertical extension is from the surface up to 5 hPa (about 35 km) with 47 levels defined in $\sigma$-pressure; with 7 levels in the planetary boundary layer, 20 in the free troposphere, and 20 in the stratosphere.

An accidental source of pollutant, such as a volcanic eruption, can be included during a simulation thanks to a special feature of the model. It requires information about the accidental emission as an input: the time, duration and place (latitude/longitude), the bottom and top plume heights, the chemical species, and the total quantity emitted.

Except for these accidental sources of emissions, other gas and aerosol pollutants emitted from natural, anthropogenic, and biomass burning sources come from inventories or parameterizations (dynamical emissions of desert dust and sea salt). In MOCAGE, with the exception of volcanic emissions, biomass burning [39], lightning $NO_x$ [42], and aircraft emissions [43], which are injected in altitude, all other chemical species sources are injected at the surface.

Both tropospheric and stratospheric air compositions are represented by using a combination of two chemical schemes. The Regional Atmospheric Chemistry Mechanism

(RACM) [44] completed with the sulfur cycle [details in [34]] is used in the troposphere and the Reactive Processes Ruling the Ozone Budget in the Stratosphere (REPROBUS) scheme is used in the stratosphere [45].

A total of 112 gaseous compounds, 379 thermal gaseous reactions, and 54 photolysis reactions are represented in MOCAGE. Primary and secondary aerosols are taken into account [34,46–48]. Primary aerosols are composed of four species: black carbon, primary organic carbon, sea salt, and desert dust. Secondary inorganic aerosols (SIA) (sulfate, nitrate, and ammonium) are implemented in MOCAGE [34]. Their concentrations are calculated with the thermodynamic equilibrium model ISORROPIA [49,50] and depend on the partition of compound concentrations between gaseous and aerosol phases and the ambient conditions (temperature and humidity). Secondary organic aerosols (SOA) are calculated from the primary carbon species [48,51].

In the CTM MOCAGE, the transport scheme is solved in two steps. Firstly, the large-scale transport (also called advection) is explicitly determined from the meteorological wind using a semi-lagrangian scheme [52]. Secondly, the subgrid scale phenomena that cannot be solved explicitly, such as convection and turbulent scattering, are parameterized. The convective transport is parameterized upon Bechtold et al. [53], while diffusion by turbulent mixing is parameterized upon Louis [54].

### 4.2. Description of the Simulations

In order to evaluate the impact of the $SO_2$ flux estimation in the plume modeling, two simulations are conducted in the Mediterranean region (from 16°N to 52°N and from 19°W to 40°E) at a spatial resolution of 0.2° longitude × 0.2° latitude. The two simulations begin on 24th December 2018 at 00 UTC and last until the end of 2018. They use the meteorological parameters from the meteorological analyses from the ARPEGE operational model (Action de Recherche Petite Échelle Grande Échelle) operated at Météo-France [55]. The anthropogenic gas emissions are from the MACCity inventory [43], while the biogenic emissions for gaseous species are from the MEGAN-MACC inventory [56]. Nitrous oxides from lightning are based on [42] and parameterized according to the meteorological forcing. Organic and black carbon aerosols are taken into account following MACCity [43]. DMS oceanic emissions are a monthly climatology [57]. Finally, the biomass burning emissions are from the daily GFAS products [58].

The first simulation, named FL, uses the $SO_2$ flux estimations retrieved by FLAME while the second simulation, named SV, uses the retrieval from SEVIRI. In the two simulations, eruptive fluxes are injected between 24th December at 10:45 UTC to 30th December at 12:30 UTC, each 15 min. In the FLAME dataset, $SO_2$ fluxes are available for only a few hours per day. In order to have a continuous emission flux in the model to simulate the emission of the volcano when not sampled by FLAME, for each day D, we performed an interpolation during the FLAME data gap. Assuming we have no other source of information, we chose to use two values for the interpolation. For the first half of the missing period, we use the last estimation available on day D and for the second half of the missing period, we use the first estimation available on day D+1. On 25th December, the presence of a meteorological cloud also masked part of the volcanic eruption for both instruments, principally for FLAME, which has no retrieved data this day. Therefore, in the same manner as for FL missing data periods, the emission flux is interpolated in the FL and SV simulations for 25th December. The corresponding emission fluxes implemented in the simulations are presented in Figure 1. For most days except 27th and 30th December, the FLAME and SEVIRI emissions at the time periods when the FLAME data are available mostly agree within their uncertainty range. On the 27th and 28th, there was a gap of data in the FLAME station located south of the crater due to a technical problem. Because the main direction of dispersion of the volcanic plume was south-westwards on the 27th and south-eastwards on the 28th, the FLAME network could only partially observe the $SO_2$ plume for these days. It leads to a significant flux underestimation compared with the space-based observations particularly on 27th December [11]. In the time period when the

interpolation is applied, there are much more important differences between FLAME and SEVIRI emissions.

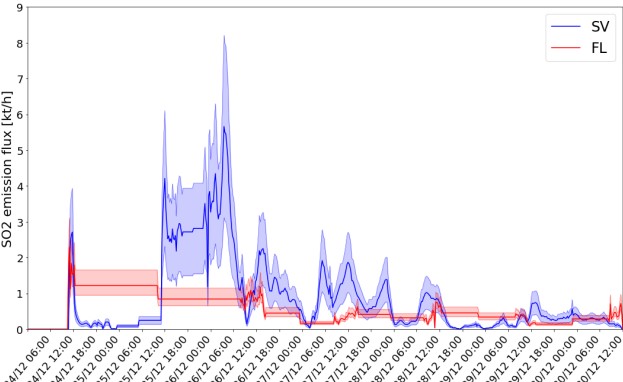

**Figure 1.** Emission fluxes as a function of time and corresponding uncertainty range, between 24 December 2018 00:00 UTC and 1 January 2019 00:00 UTC (in kt/h), used as input in FL (red) and SV (blue) simulations.

The volcanic plume is injected in altitude in MOCAGE, following an umbrella profile [1], using the top heights retrieved by both the SEVIRI instrument and the visible camera (Table 1). On 24th December, we only have one estimation of the plume height using the dark pixel algorithm [11], from the SEVIRI instrument, which corresponds to the maximal height measured this day. In the simulations, we use a maximum plume height of 8 km only at the peak of the eruption (when the emission flux is greater than 350 kg/s or 1.26 kt/h). When the emission flux is below this threshold, we set the maximum plume height to 5 km. This corresponds to times after 13:00 UTC and 12:00 UTC in FL and SV, respectively. This 5 km value is chosen from Figure 5 in Corradini et al. [11], which highlights that, after the peak of the eruption on 24th December, the volcanic plume was injected at an altitude around from 4 to 6 km. From 25th to 30th December, the volcanic plume was too transparent to apply the dark pixel procedure, so the visible camera was exploited. On 25th December, due to the data gap in the visible camera, we have no estimate of the plume height, so we use the same as on 26th December.

**Table 1.** Plume heights used (in kilometers) between 24th December at 10:45 UTC and 30th December at 12:30 UTC in FL and SV simulations; based on [11].

| Day | Plume Height—FL | Plume Height—SV |
|-----|-----------------|-----------------|
| 24 | 8.0, 5.0 after 13:00 UTC | 8.0, 5.0 after 12:00 UTC |
| 25 | 4.0 | 4.0 |
| 26 | 4.0 | 4.0 |
| 27 | 4.5 | 4.5 |
| 28 | 5.5 | 5.5 |
| 29 | 4.5 | 4.5 |
| 30 | 4.5 | 4.5 |

Except for the eruptive phase, the volcanic emission flux is considered as passive degassing (before 10:45 UTC on 24th December and after 12:30 UTC on 30th December). In both simulations, we use the estimation of passive emissions from Mount Etna for 2015 for these periods, from the inventory of Carn et al. [17]. The year 2015 is chosen as the most recent year available in the inventory. The plume height is injected between 3300 (altitude of Mount Etna) and 3400 m, corresponding to a reasonable altitude for a plume from passive emissions at Mount Etna [17].

## 5. Results

In order to evaluate the representation of the volcanic plume in the two simulations, we compare the $SO_2$ total column simulated with those observed by the spaceborne instruments IASI_A, IASI_B, OMI, TROPOMI_OP, and TROPOMI_MPIC. Each day, we have between two and five observation swaths available over the Mediterranean region (see Figure 2). For the model–observation comparison, we did not choose to interpolate the simulations outputs at the observation points since the satellite $SO_2$ column estimations often cover only part of the volcanic plume or are noisy (because of detection limits, clouds, or the location of the swath). Note also that TROPOMI and IASI have a finer resolution than the simulations, thus meaning that the model cannot provide information at the same scale as these two products. Instead, we chose to perform a semi-quantitative assessment based on comparisons of the maps of $SO_2$ total columns keeping the model and satellite products at their native resolutions. Since the simulations provide a full $SO_2$ map over the Mediterranean region, it is possible to perform a detailed analysis of the simulated plume transport. In practice, because the model provides hourly outputs, the simulation time used for the comparison is the closest hour to the overpass time of each instrument.

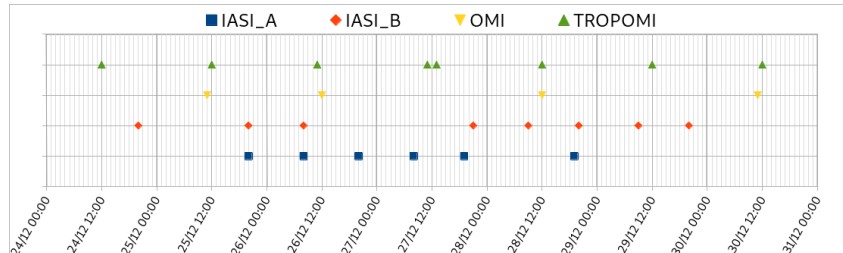

**Figure 2.** Crossing time over the Mediterranean region for each instrument.

In order to be clear in the comparison against the observations, all parts of the volcanic plume are numbered and the numbering is reset each day. Indeed, the comparison is performed with multiple sensors, not available at the same time, so the numbering helps the comparison but is not meant to follow the full temporal evolution of the plume between 24th and 30th December. Due to the time of the beginning of the eruption on 24th December and due to the presence of a meteorological cloud over the Mediterranean region on 25th December, we lack observational data for the evaluation at the early stage of the eruption. Hence, in this work, the analysis is performed in the period between 26th and 30th December.

### 5.1. 26th December

Five satellite swaths are available for the comparison on 26th December: three swaths from IR instruments and two from UV instruments.

Beginning with the comparison against IR instruments (IASI_A and IASI_B), Figure 3 presents the total column of $SO_2$ from the simulations (FL for the left column and SV for the middle column) and the observations (right column) on 26th December. In the morning, at 08:00 UTC (Figure 3A), the volcanic plume location was consistent between the two simulations and with the observations. However, concerning the quantity of $SO_2$, there are large differences between FL and SV in Plumes 1–2–3, with higher concentrations of $SO_2$ modeled in the SV simulation. This result is coherent with the flux emitted by the volcano from 25th December at 12 UTC to the 26th early morning: around 3.2 kt/h of $SO_2$ emitted in SV but four times less, 0.8 kt/h, in FL (Figure 1). Compared to IASI, the FL simulation provides better quantitative results even taking into account SEVERI's large uncertainty (45%, see Figure 1). In the evening, the instrument IASI_A provides observations at 20:00 UTC (Figure 3B), but only very close to the vent. Plume 4 seems more realistic in the SV simulation, where $SO_2$ concentrations within the plume are higher: at around 12 and 15 DU (Dobson Units). However, when performing the comparison between IASI products and the simulations, one has to keep in mind that its sensitivity below 5 km

is low. Because concentrations in the plume on 26th December correspond to emissions up to 4 km, the IASI columns may be underestimated.

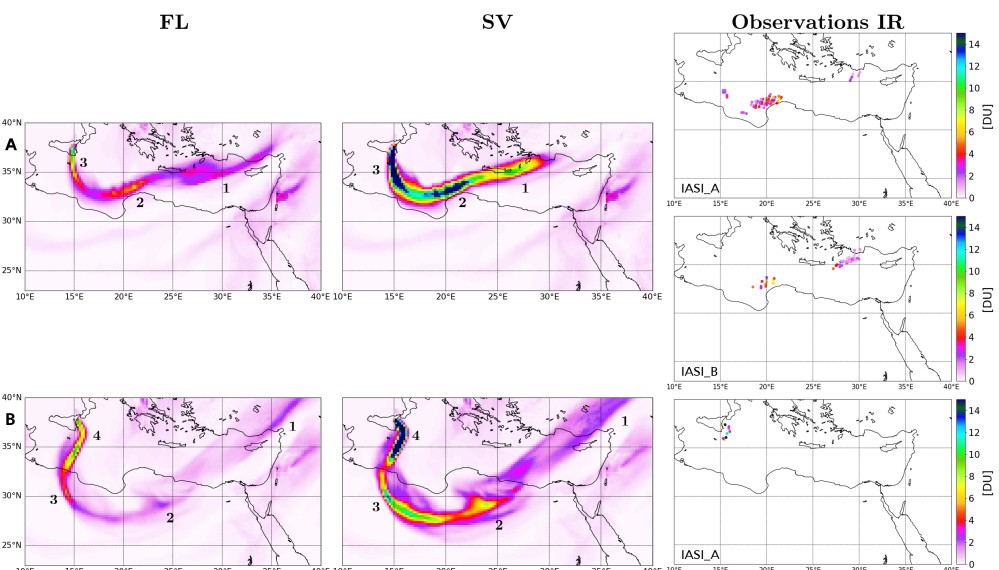

**Figure 3.** Total column of $SO_2$ in Dobson Unit (DU) simulated in FL (**left panels**) and SV (**middle panels**) and observed using the IR instruments on 26th December (**A**) at 08:00 UTC (IASI_A = **top right panel** and IASI_B = **middle right panel**) and (**B**) at 20:00 UTC (IASI_A = **bottom-right panel**).

This is why UV instruments are used as well for the evaluation (Figure 4). We note in Figure 4 that almost all the volcanic plume is detected (especially by the TROPOMI instrument). This allows us to evaluate in more detail than with IASI the transport of the volcanic plume in the FL and SV simulations. We can conclude that the transport is well-simulated by the model in both simulations. One difference in the plume transport is the most eastern part of Plume 1 located north-west of Cyprus that is hardly visible in the SV simulation compared to the observations. This corresponds to the low emissions at the end of the day on 24th December in the SEVIRI emissions that lead to small quantities of $SO_2$ there. On the contrary, this part of Plume 1 is correctly represented in the FL simulation where the flux emissions interpolated on 24th December leads to greater amounts of $SO_2$.

Concerning the quantity of volcanic $SO_2$, we first perform a comparison against TROPOMI_OP and TROPOMI_MPIC at 11:00 UTC (Figure 4A). The $SO_2$ columns simulated in FL are lower than the two datasets of observations, with the exception of Plume 3. In the SV simulation, the concentrations of volcanic $SO_2$ are consistent for Plumes 1 and 2 but slightly higher. For example, for Plume 2, the maximum of volcanic $SO_2$ simulated is 13 DU against 11 DU for the estimations of the two TROPOMI products. However, for Plume 3, the columns of $SO_2$ simulated in SV are much higher than the observations. This could be due to an overestimation of the $SO_2$ emission flux by SEVIRI due to ash.

Secondly, we compare the simulations against the OMI estimates (Figure 4B) at 12:00 UTC. Here, we note that Plume 2 and 3 in the SV simulation are consistent with the observations in location and in quantity of $SO_2$ within the plume but with a slight overestimation in Plume 3. Against OMI, the modeling of the volcanic plume on 26th December is overall better in the SV simulation. The discrepancies between the TROPOMI and OMI estimates are mainly due to the different retrieval algorithms used and to the differences of resolution between the two instruments.

Note that the plume associated with the eruption reaches high values of concentrations with $SO_2$ columns values > 12 DU. Compared to the recent climatological values produced by [59] from OMI and OMPS (Ozone Mapping and Profiler Suite) spaceborne instruments, these concentrations are much higher than the background columns in the Mediterranean basin ( 0.1 DU and 0.25 DU for OMI and OMPS, respectively). They are also stronger than

the climatology in the vicinity of Mount Etna where [59] show a local maximum of $SO_2$ of 0.2 DU and 0.4 DU for OMI and OMPS, respectively. The emissions from this eruption are therefore much larger than the average Mount Etna passive degassing.

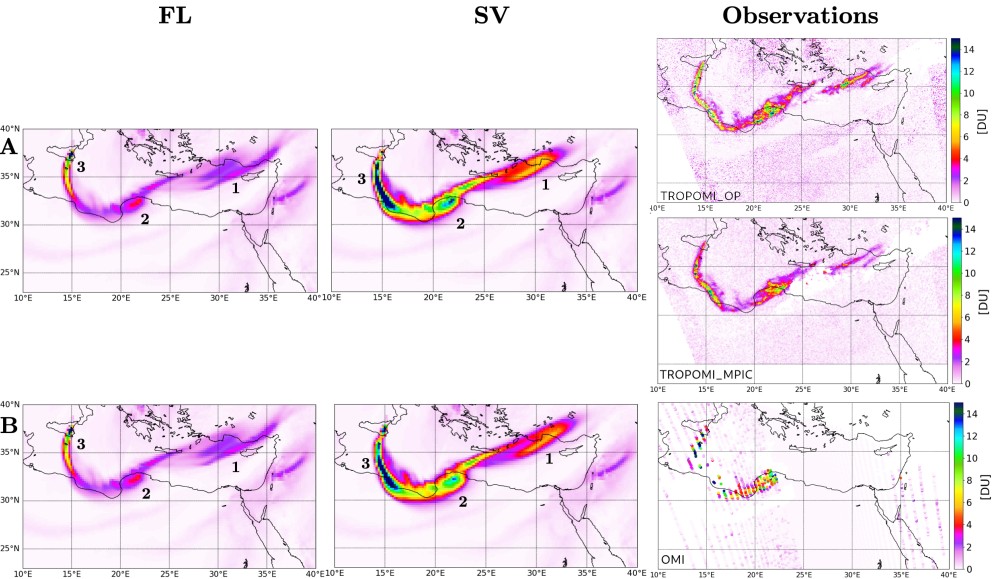

**Figure 4.** Total column of $SO_2$ in Dobson Unit (DU) simulated in FL (**left panels**) and SV (**middle panels**) and observed using the UV instruments (**right panels**) on 26th December at (**A**) 11:00 UTC (TROPOMI_OP = **top right panel** and TROPOMI_MPIC = **middle right panel**) and (**B**) 12:00 UTC. (OMI = **bottom-right panel**).

To conclude, the plume is generally well located in the two simulations meaning that the top height of the plume used in the simulation is reliable from the night of the 24th to midday on 26th December. The comparison with the IR $SO_2$ concentrations tends to show that FLAME emissions are quantitatively better than SEVIRI's. On the contrary, even if there are differences in the $SO_2$ concentrations between TROPOMI_OP, TROMPOMI_MPIC, and OMI, the comparison indicates that the SV simulation is closer to the observations. This can be explained by the fact that the IR columns are underestimated because of their low sensitivity below 5 km.

### 5.2. 27th December

On 27th December, we have a total of five crossings of satellite instruments available, but, here, we only select the two with enough observation points. Therefore, we only show the comparison against the UV instrument TROPOMI at 11:00 UTC and 13:00 UTC.

Figure 5 presents the total column of $SO_2$ simulated in FL and SV and the observations of the two datasets TROPOMI_OP and TROPOMI_MPIC. Even if Plume 1 is on the edge of the TROPOMI swath and not totally detected, the transport of the plume is correct in the simulations with a location of the plume consistent with the complex shape showed in the observations. However, the spatial variability within the volcanic plume is not as fine as in the TROPOMI observations, especially for Plumes 3 and 2. The modeled plume is smoother and more spread out as expected because the model resolution is coarser ($0.2° \times 0.2°$) than the TROPOMI pixels.

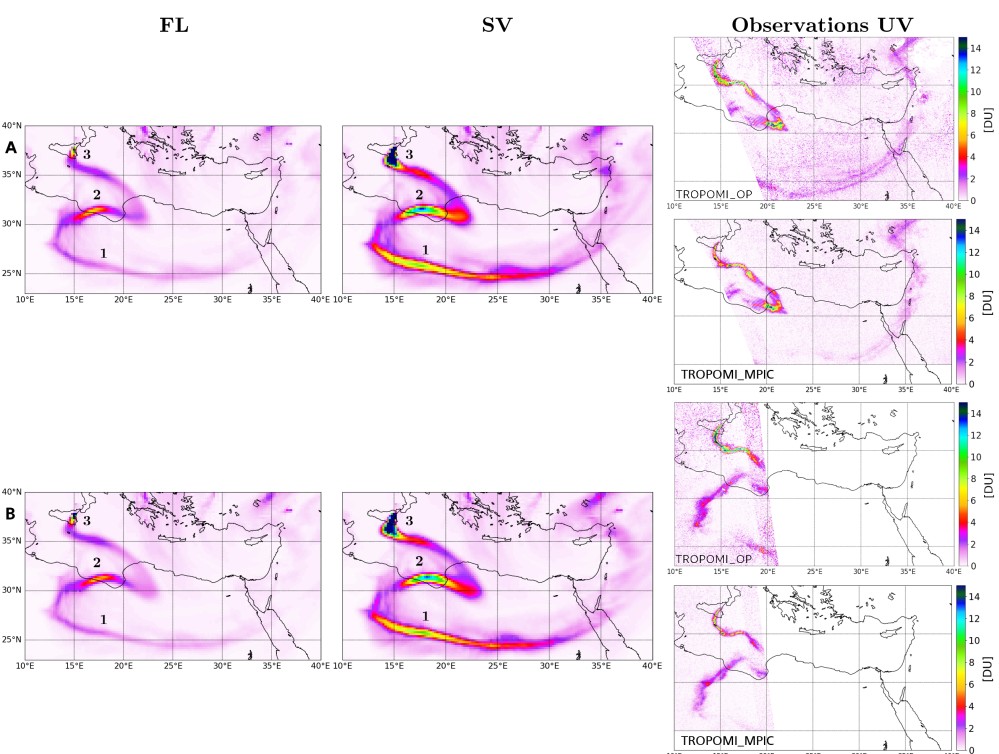

**Figure 5.** Total column of SO$_2$ in Dobson Unit (DU) simulated in FL (**left panels**) and SV (**middle panels**) and observed using the UV instruments (**right panels**) on 27th December at (**A**) 11:00 UTC (TROPOMI_OP = **top right panel** and TROPOMI_MPI = second to top right panel) and (**B**) 13:00 UTC (TROPOMI_OP = second to bottom-right panel and TROPOMI_MPI = **bottom-right panel**).

In terms of the quantity of SO$_2$ within the plume, the FL simulation is lower than the SV simulation. For Plume 3 at 11 UTC and 13 UTC, the maximum amount of SO$_2$ simulated is higher in the SV simulation than in FL due to the lower emissions in the FL simulation on 27th December from 00 UTC. For Plume 3, both simulations have a higher column ($\geq$14 DU) than the maximum of SO$_2$ observed by the product TROPOMI_OP and TROPOMI_MPIC, with 10 and 8 DU, respectively, and only a very local maximum of 14 DU. These differences could be due to too high emissions in FL and SV or to the presence of volcanic ash close to the vent. Even a small quantity of ash can mask part of the SO$_2$ signal and affect the retrieval from TROPOMI leading to an underestimation [60,61]. For Plume 2, which corresponds to emissions on 26th December late afternoon and evening in the simulations, there is a shift of the location of the maximum meaning that either the model transport is too strong or the variability of the emissions in both FL and SV are not fully right. Apart from this shift of the position of the maximum, the SV SO$_2$ values are closer to observations than FL. For Plume 1, the SV simulation exhibits concentrations of SO$_2$ much higher than FL. Even if this part of the plume is only partly detected using TROPOMI, the SV simulation clearly overestimates the SO$_2$ column. Plume 1 corresponds to the emissions on 26th December before 06 UTC that were already showed to be too high in the SV simulation (corresponding to Plume 3 in Figure 4).

### 5.3. 28th December

For 28th December, we have observations both using IR and UV instruments available (Figures 6 and 7).

We can see that the transport of the volcanic plume is even more complex than for previous days, but the model mostly succeeds to represent it. We note on the total column of SO$_2$, and particularly on the SV simulation, the presence of several bands of volcanic plume spread from west to east between 31 and 36°N. This is due to a strong vertical wind shear that changes the direction of the plume depending on the altitude. The IR

instrument IASI_B does not properly detect this complex structure, only slightly at 09 UTC because of the limited number of pixels available. The emissions on 28th December are up to 5.5 km altitude and IASI_B mainly captures the parts of the plume that are located above 5 km because of its low sensitivity below this altitude. This can also explain why the estimation of the concentrations of $SO_2$ retrieved using IASI_B is lower than in both the FL and SV simulations (Figure 6). However, the difference between the simulations and the observations is not as strong as on the 26th since the volcanic plume was emitted at a higher altitude (5.5 km) on the 28th compared to the 26th (4 km).

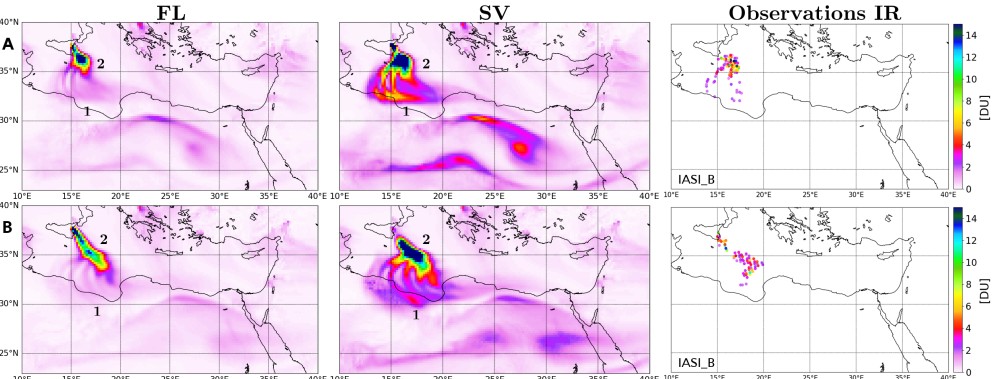

**Figure 6.** Total column of $SO_2$ in Dobson Unit (DU) simulated in FL (**left panels**) and SV (**middle panels**) and observed by the IR IASI_B instrument (**right panels**) on 28th December at (**A**) 09:00 UTC and (**B**) 20:00 UTC.

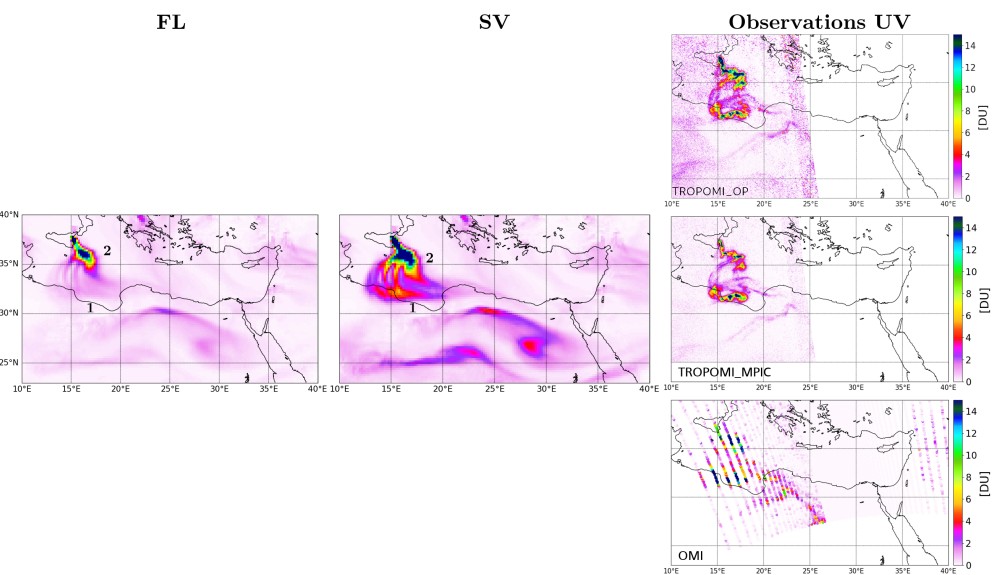

**Figure 7.** Total column of $SO_2$ in Dobson Unit (DU) simulated in FL (**left panels**) and SV (**middle panels**) and observed using the UV instruments TROPOMI_OP, TROPOMI_MPIC, and OMI (**right panels**), 28th December at 12:00 UTC.

The estimations of the amount of $SO_2$ from the three UV instruments are consistent but with some differences (Figure 7). As noted before, the columns of $SO_2$ estimated from OMI are higher than from TROPOMI. There are also small differences in the estimates from the two TROPOMI products. The quantity of $SO_2$ observed by TROPOMI_OP is a bit higher in Plume 2.

Compared to the UV instruments, the SV simulation (Figure 7) is generally closer than the FL simulation. We can see a clear signature of Plume 1 (Libyan coast) in the SV simulation that is also detected by all the UV instruments. The column of $SO_2$ in FL is

much lower (maximum around 1–2 DU) with respect to the SV simulation (maximum around 7 DU) and the observations (maximum above 12 DU). This result is explained by the stronger flux of emissions in the SV simulation (maximum of 1.4 kt/h of $SO_2$ emitted) compared to the FL simulation (maximum of 0.6 kt/h), on 27th December from 18 UTC (see Figure 1). For Plume 2, the SV simulation compares better to OMI and TROPOMI_OP observations with a large area of high concentrations southeast of Mount Etna while the FL simulation is closer to TROPOMI_MPIC. Still, the SV simulation tends to slightly overestimate the $SO_2$ columns.

### 5.4. 29th December

On 29th December, the IR and UV observations are available and are discussed separately.

Figure 8 presents the comparison between the $SO_2$ total column simulated and observed using IASI_B at 20:00 UTC. We notice differences in the representation of the volcanic plume location between the two simulations. In the FL simulation, Plumes 1 and 2 are linked to each other. On the contrary, in the SV simulation, Plumes 1 and 2 are clearly separated. This is due to much lower $SO_2$ emissions estimated from the SEVIRI instrument compared to FLAME between 28th December at 14:00 UTC and the 29th at 9:00 UTC (see Figure 1). However, apart from this period, the emission fluxes injected in the SV simulation are higher than in the FL simulation. This explains the differences for the maxima of volcanic $SO_2$ concentrations within Plumes 1 and 2. By comparing the column of $SO_2$ modeled using MOCAGE against the IASI_B retrievals, we can see that, near the source (Plume 2), the best simulation is SV (from 8 to 10 DU). It is not possible to perform an assessment of Plume 1 because of the lack of IASI_B data.

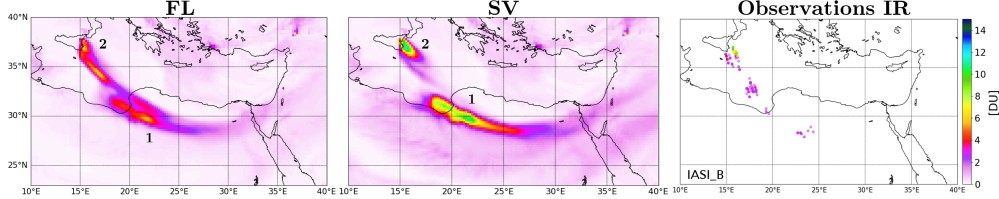

**Figure 8.** Total column of $SO_2$ in Dobson Unit (DU) simulated in FL (**left panels**) and SV (**middle panels**) and observed using the IR IASI_B instrument (**right panels**) on 29th December at 20:00 UTC.

The comparison against TROPOMI_OP and TROPOMI_MPIC, at 12:00 UTC (Figure 9) highlights that the model is not fully representing the complex transport of the volcanic plume. In both the SV and FL simulations, Plume 1 is broader than in the observations and does not clearly show the small-scale features of the plume exhibited using TROPOMI_MPIC and TROPOMI_OP around (17°E, 33°N) linked to the wind variability. This can be related to the relatively coarse vertical and horizontal resolutions of the model simulation. Moreover, both simulations missed the southern part of the volcanic plume. The model transports emissions eastwards from Plume 1 and not toward the south. This is probably because of the emissions not being injected at the right altitude in the model being linked to uncertainties in the estimate of the top plume height from the cameras.

Concerning the quantity of $SO_2$ within the plume, the SV simulation and the UV observations are similar for Plume 2 (Figure 9). For Plume 1, there are differences in the quantity of $SO_2$ between TROPOMI_OP and TROPOMI_MPIC, with TROPOMI_MPIC being lower as already found in the comparison for previous days. The FL simulation, which is between 4 and 8 DU, is closer to the retrievals from TROPOMI_MPIC and TROPOMI_OP while the SV simulation is too high (between 7 and 13 DU). Plume 1 on 29th December (Figure 9) corresponds to Plume 2 on 28th December (Figure 7), which was higher in the SV simulation compared to TROPOMI estimates (see previous section).

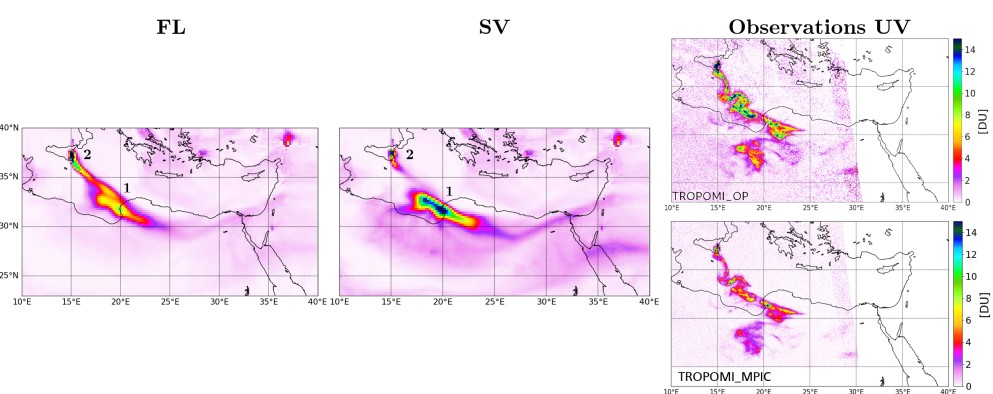

**Figure 9.** Total column of SO$_2$ in Dobson Unit (DU) simulated in FL (**left panels**) and SV (**middle panels**) and observed using the UV instruments TROPOMI_OP and TROPOMI_MPIC (**right panels**) on 29th December at 12:00 UTC.

### 5.5. 30th December

On 30th December, the last day of the eruption, we only have observations from UV instruments (Figure 10). As for previous days, OMI provides greater SO$_2$ columns than TROPOMI products, while TROPOMI_OP is a bit higher than TROPOMI_MPIC. There is an overall good consistency between the three products on the location of maxima and minima except on the southern part of the plume (around 26°E 26°N). Compared to the simulations, the TROPOMI products provide a very detailed description of the plume features that cannot be captured by the model because of its resolution. Still, we note the good location of Plumes 1 and 3 in FL and SV simulations. Plume 2 is largely underestimated by the model, especially in the SV simulation in which Plume 2 is not visible. This "missing" plume is associated with the very low emissions of 29th December in the SV simulation.

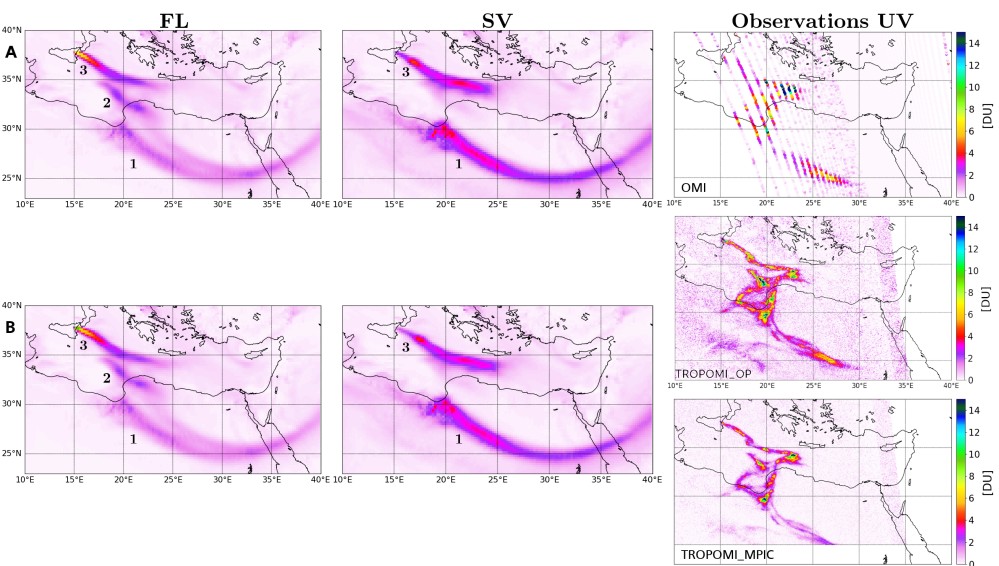

**Figure 10.** Total column of SO$_2$ in Dobson Unit (DU) simulated in FL (**left panels**) and SV (**middle panels**) and observed using the UV instruments on 30th December at (**A**) 11:00 UTC (OMI = **top right panel**) and (**B**) 12:00 UTC (TROPOMI_OP = **middle right panel** and TROPOMI_MPIC = **bottom-right panel**).

The concentrations of SO$_2$ are generally lower in both simulations compared to those observed using the different instruments with the SV simulation being generally higher than FL except close to the vent and for Plume 2. Therefore, the emission flux is likely under-estimated, both in the flux estimations based on FLAME and SEVIRI instruments between 29th December at 15 UTC and 30th December at 06 UTC.

## 6. Discussion

Following the day-by-day analysis, we summarize and discuss the results here in a more general manner.

For this case study, the comparison between the IASI, OMI, and TROPOMI SO$_2$ columns shows a very good agreement on the location of the plume and an overall consistency for the SO$_2$ quantity but with several noticeable differences. Firstly, the IR instruments (IASI_A and IASI_B) provide less observational data of the plume and retrieve lower quantities of volcanic SO$_2$ for this case study compared to the UV instruments. Given the low sensitivity of the IR instruments in the altitudes below 5 km, part of the volcanic plume concentration is not detected, especially on the days when the injection heights of the volcanic plume were below 5 km. Therefore, before using IASI for the evaluation of the simulated SO$_2$ concentrations, one has to ensure that the volcanic plume is high enough. Secondly, for the UV instruments, the use of different algorithms between OMI and TROPOMI leads to differences in the retrieved columns. The instrument OMI, using a PCA algorithm, retrieves higher amounts of SO$_2$ compared to TROPOMI, using a DOAS algorithm. Note that with the finest spatial resolution, TROPOMI provides more detailed information on the fine scale features within the plume than OMI and IASI. Finally, even with the same instrument (TROPOMI), the use of similar but slightly different algorithms can generate differences in the retrieved columns. Here, the observational data from the TROPOMI_OP product are generally a bit higher than those from the TROPOMI_MPIC product, but in some local places the reach a factor of 2.

Knowing this, we now discuss the evaluation of the two simulations compared to the observations starting with the representation of the volcanic plume transport. The location of the volcanic plume is similar in the two simulations because they use the same altitude of injection of the emissions. The location of the volcanic plume in both simulations is very consistent with the observations from all the instruments even far from the vent with only a few exceptions. This indicates that the estimation of the emission maximum height from the observations used as input information in the model is good overall. The consistency between the simulations and the observations regarding the plume location also shows that the horizontal and vertical resolutions of the model are fine enough to, in general, correctly describe the transport of the plume even when wind shear occurs. However, some of the complex structures of the plume provided by the TROPOMI fine resolution product are not fully represented in the simulations within the plume. This is partly due to the relatively coarse vertical resolution of MOCAGE in the mid and upper troposphere (700–800 m) limiting the exact and fine representation of the impact of wind shears. Furthermore, the model provides more spread and smoother plumes because of its horizontal resolution (coarser than TROPOMI) and the diffusivity of its transport scheme. Furthermore, on a few occasions, some parts of the volcanic plume were detected using the spaceborne instruments and missing in the simulations. It was due to a very low flux of emission used in the simulations. One way to better simulate the complex features of the plume would be to increase the number of vertical levels (i.e., finer resolution) together with a finer horizontal resolution.

Since the transport of the plume is generally well represented in the model, it causes it to be possible to discuss the representation of the quantity of SO$_2$ within the plume modeled in the two simulations. The two simulations provide different concentrations of SO$_2$ within the plume, with generally stronger concentrations in the SV simulation. This is because the fluxes injected into the FL simulation are mostly smaller than those injected into SV. As expected, there is a strong sensitivity of the concentrations within the volcanic plumes to the emission fluxes. This was also shown in [62] for another Mount Etna eruption case study. The comparison of the simulations with the satellite-based estimates of SO$_2$ columns shows that the SV simulation is slightly better than FL, but this is not always the case. This highlights that there are uncertainties on the estimation of the emissions fluxes from the two instruments (FLAME and SEVIRI). Part of the uncertainty comes from the observations and methods used to retrieve the emission fluxes. Moreover, the presence of

ash can lead to an overestimation of the emission flux from SEVIRI even after the correction is applied by the algorithm. For FLAME, the impact of ash is an underestimation. Another part of the uncertainty is linked to the data availability. The emission fluxes retrieved from the SEVIRI instrument are available during the whole period of the eruption except over a 12-h time period on 25th December, unlike those from FLAME, which are available only a few hours per day and not every day. To ensure the continuity of the emission flux in the model, we completed the missing part of the FLAME emission fluxes using an interpolation that generally leads to less realistic results in the FL simulation than in SV. These two datasets could be combined to provide a better estimate of the $SO_2$ fluxes. The FL estimates could be used at the time periods when it is available because it has a lower uncertainty when measuring in good conditions and SV could be used otherwise.

Even if they are slightly better, the SV simulation still shows significant differences in the concentrations in several plumes near the vent compared to the observations with the $SO_2$ in the plume being higher in SV. Apart from the general emission flux uncertainty from the retrieval method and observations, the presence of ash, even in small quantities, may provide an under-estimation in the retrieval of the column concentrations using spaceborne instruments [27,60,61]. In contrast, further away from the vent, the comparison between the simulated (in SV) and observed concentrations of $SO_2$ are more coherent but not for all parts of the plume. We also find that the uncertainty on the modeled plume concentrations linked to the emission fluxes is particularly important in both simulations on the first days of the eruption. The presence of a meteorological cloud on 25th December in the morning, masking a large part of the plume (not shown here), leads to a gap in the emission estimation and to large uncertainties because the emissions had to be interpolated.

Another important point to note is that the differences between the $SO_2$ columns from IASI, OMI, TROPOMI_OP, and TROPOMI_MPIC can be nearly as large as those between the SV and FL simulations in some parts of the plumes (e.g., Plume 1 on 26th December, tail of Plume 1 on 30th December, etc.). This is linked to the fact that the spaceborne $SO_2$ column products have uncertainties. These uncertainties need to be taken into account when evaluating the model simulations against these products.

In this study, we have used a very simple semi-quantitative method to compare the simulations to the satellite-derived $SO_2$ columns, i.e., on the basis of maps of the column concentrations at the closest time. We did not choose to interpolate the simulation outputs at the observation points because the satellite $SO_2$ pixels do not always cover the whole volcanic plume (because of detection limits, clouds, or the location of the swath). This also allowed us to obtain the full picture from the model of the plume location and spread. A better method to perform a quantitative and fairer comparison would be to use information on the averaging kernels (e.g., Eskes and Boersma [63]). The reason they were not applied to the results of the MOCAGE model is because they were only available for one product (TROPOMI_OP). In order to analyze the impact of applying the averaging kernels on the model results, we performed this operation on the MOCAGE simulations based on TROPOMI_OP dataset. We used the product that assumes a $SO_2$ plume height at 7 km because it is the closest to the 5 km average height considered in this case study. We illustrate the results for two days showing different behaviors. Figure 11 presents results of this comparison for 26th December at 11:00 UTC. The FL and SV panels of this figure can be compared to Figure 4 in order to see the impact of applying averaging kernels to the model results. We can note that the plume location remains the same but the values are lowered when applying the averaging kernels to the simulated data. We can also see that the observation values are slightly lower when considered at 7 km than at 5 km. The histogram presented in Figure 11 compares the distribution of the total column values between TROPOMI_OP and the FL and SV experiments. For this day, the SV simulation effectively reproduces the observed values apart from the values above 12 DU that are missing. This can be explained by the fact that, in the observations, these values correspond to small areas within the plume with high values that are hardly represented using MOCAGE. The FL simulation has too low values of the $SO_2$ total column compared to the observations and

the SV simulation provides better results as shown by the histogram. This is consistent with the previous analysis performed in Section 5.1. Figure 12 presents results for 28th December at 12:00 UTC. As for 26th December at 11:00 UTC, applying the averaging kernels at 7 km leads to lower values in both simulations compared to Figure 7. For 28th December at 12:00 UTC, the histogram confirms that the FL simulation provides lower concentrations than the SV simulation and that there is an overestimation of SV values compared to TROPOMI_OP, which is consistent with the analysis in Section 5.3.

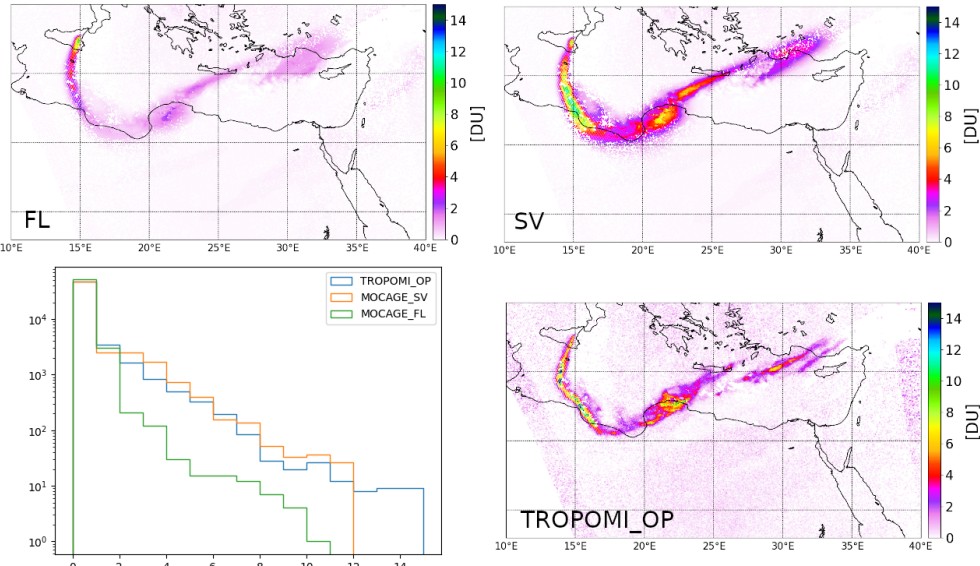

**Figure 11.** Total column of SO$_2$ in Dobson Unit (DU) from FL (**upper left**) and SV (**upper right**) simulations with application of the TROPOMI_OP averaging kernel and observed using the TROPOMI instrument (TROPOMI_OP at 7 km, **bottom right**) on 26th December at 11:00 UTC. The **bottom-left** panel represents the histogram of values from FL, SV and TROPOMI_OP data.

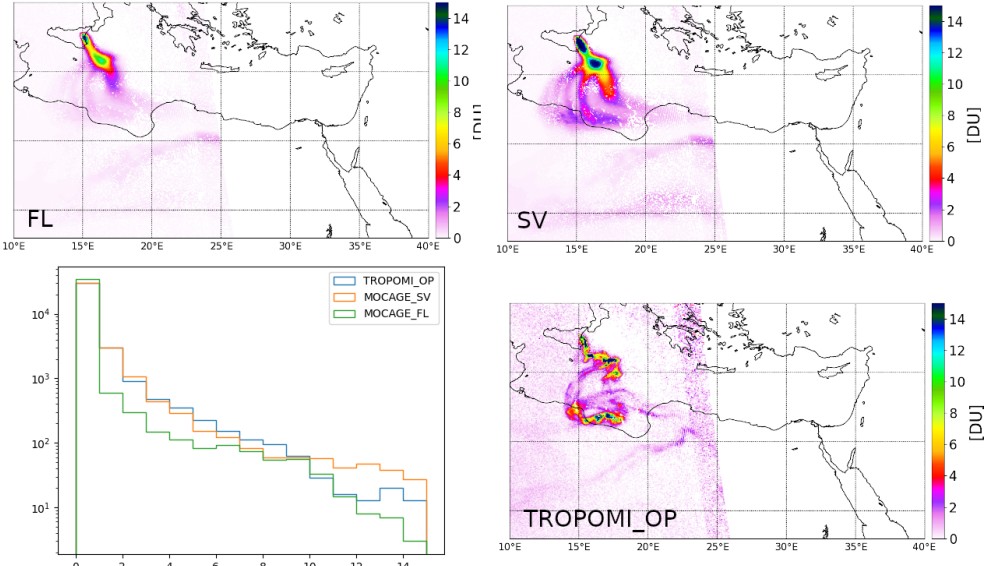

**Figure 12.** Total column of SO$_2$ in Dobson Unit (DU) from FL (**upper left**) and SV (**upper right**) simulations with application of the TROPOMI_OP averaging kernels and observed using the TROPOMI instrument (TROPOMI_OP at 7 km, **bottom right**) on 28th December at 12:00 UTC. The **bottom-left** panel represents the histogram of values from FL, SV, and TROPOMI_OP data.

### 7. Conclusions

Two simulations, using two different estimations of the emission flux of $SO_2$ from the Mount Etna eruption of Christmas 2018, are conducted using the MOCAGE model and compared against multiple spaceborne observations. One estimation is from the ground-based network FLAME and the other is from the spaceborne instrument SEVIRI. The general aim of this paper is to study the impact of the use of these two estimations on the modeling of the volcanic plume over the whole period of the eruption (7 days from 24th December) at the regional scale over the Mediterranean basin. To evaluate the simulations, we compare the modeled column of $SO_2$ against $SO_2$ retrievals from the IASI, OMI, and TROPOMI instruments between 26th and 30th December. In this section, we summarize the main results of the comparison between the model and the observation instruments.

Concerning the modeling of the volcanic plume in MOCAGE, with the same altitude of injection of the emissions in both simulations, we note that the model correctly represents the transport and location of the plume during the whole eruptive period and is consistent with the observations. Similarly to [62], we confirm that the emission height parameter is fundamental for plume modeling. However, due to generally higher emission fluxes injected in the SV simulation compared to FL, some differences are shown regarding the quantity of $SO_2$ modeled within the plume between the two simulations. The concentrations of $SO_2$ in the SV simulation are generally stronger and are in slightly better agreement with the observation instruments, but not always. As discussed, this can be explained by several factors: the uncertainties in the estimation of the emission fluxes and the uncertainties linked to the data availability. The SEVIRI instrument is almost always available, except for a 12-hour time period on 25th December, when a meteorological cloud masked the volcanic plume and FLAME is only available for a few hours a day and not every day. The uncertainties on the emission fluxes from both instruments are generally large: from −22 to +36% for FLAME and ±45% for SEVIRI. In particular, the presence of ash can lead to an underestimation (overestimation) of the emission fluxes for the ground-based instrument FLAME (the spaceborne instrument SEVIRI).

Overall, all the $SO_2$ column retrievals from the different observation instruments that were used for the evaluation of the simulations are consistent between one another, but differences can be noticed. The IR instruments (IASI_A and IASI_B) provide less data for this case study, due to their smaller sensitivity in the lower troposphere. On the contrary, more data are available with the UV instruments (OMI and TROPOMI), but the use of two different algorithms leads to small differences in the $SO_2$ retrievals. This highlights the uncertainty of the observational data available to validate the model simulations.

The analysis of the simulation results is based on a simple semi-quantitative method for the model—observation comparisons. To perform a more quantitative analysis, it would be useful to have the averaging kernels for each observation product of the $SO_2$ column and to apply them to the simulated $SO_2$ concentrations, because they take into account the characteristics of each instrument. This could be particularly important for the comparison with IR instruments that have a low sensitivity below 5 km altitude. For a fair comparison between the $SO_2$ columns from different satellite observations, the averaging kernels were not applied because they were only available for TROPOMI_OP and its model results. However, we assessed the impact of using them. The distribution of the concentration values using averaging kernels provides an objective comparison between the simulations and the observations. The results with the averaging kernels are consistent with those of the semi-quantitative method.

A promising method to improve the simulation of volcanic plumes is to perform a data assimilation of the quantity of $SO_2$ retrieved using the spaceborne instruments data into a model. This is currently under development and evaluation in MOCAGE. In the case of advanced data assimilation methods, both the observation and the model errors can be taken into account to provide an optimal model estimate of the $SO_2$ concentration where the observations are available (e.g., [64]).

**Author Contributions:** Conceptualisation: C.L., J.G. and V.M.; Model simulations: C.L. and J.G.; Simulation post-processing: C.L., M.B. and J.G.; Volcanic flux data: G.S., S.C. and L.G.; Satellite-derived concentrations: S.W., T.W., N.T. and H.B.; writing—original draft preparation: C.L., V.M. and J.G.; Writing—review and editing, all authors. All authors have read and agreed to the published version of the manuscript.

**Funding:** This research received no external funding.

**Data Availability Statement:** Estimations of the emission flux using FLAME and SEVIRI were provided to us by personal communication from researchers of the INGV. Those data are available upon request at the corresponding co-authors. The retrievals of the $SO_2$ column from TROPOMI were provided to us by personal communication from researchers of the BIRA and MPIC. The data from the MPIC are available upon request from the corresponding co-authors and data from the BIRA correspond to the operational product TROPOMI $SO_2$ column, which is public and available on the Copernicus Sentinel-5P data center (https://s5phub.copernicus.eu (accessed on 11 March 2020)). IASI is a joint mission of EUMETSAT and the Centre National d'Etudes Spatiales (CNES, France). The authors acknowledge the AERIS data infrastructure for providing access to the IASI data in this study, ULB-LATMOS for the development of the retrieval algorithms, and Eumetsat/AC SAF for CO/O3 data production.

**Acknowledgments:** We would like to acknowledge the Université Paul Sabatier Toulouse III for supporting the doctoral research contract of Claire Lamotte and thank Météo-France for hosting Claire Lamotte's doctoral research at the Centre National de Recherches Météorologiques.

**Conflicts of Interest:** The authors declare no conflict of interest.

## Appendix A. Properties of the TROPOMI_MPIC Algorithm and Comparison with the TROPOMI_OP Algorithm

While both TROPOMI_OP and TROPOMI_MPIC employ three DOAS fit windows for different $SO_2$ loads, TROPOMI_MPIC omits the UV fit range at a large wavelengths (w3, 360–390 nm) of TROPOMI_OP and adds a fit range at UVs between w1 and w2 of TROPOMI_OP (see Table A1 and [27,65]). Furthermore, TROPOMI_MPIC applies an interpolation between the fit ranges, which leads to a smooth transition between the corresponding fit results. It should be noted that the omission of the UV fit range at a large wavelength (360–390 nm) can lead to a slight underestimation of the $SO_2$ column for very high columns of more than 250 DU. Another difference between both algorithms is the AMF calculation. TROPOMI_MPIC uses a different radiative transfer model (MCArtim [66]) and calculates the AMF directly for a Gaussian plume at 5 km, while for the TROPOMI_OP, a look-up-table is employed.

**Table A1.** Overview over the $SO_2$ SCD calculation schemes of the two products TROPOMI_OP and TROPOMI_MPIC used in the study. Detailed information can be found in [27] (TROPOMI_OP) and [65] (TROPOMI_MPIC).

| | TROPOMI_OP | | TROPOMI_MPIC | |
|---|---|---|---|---|
| | wavelength (nm) | DU threshold | wavelength | DU threshold |
| w1 | 312–326 | SCD < 15 | 312–324 | SCD < 11.5 |
| transition w1/2 | - | - | interpolation | 11.5 < SCD < 30 |
| w2 | 325–335 | 15 < SCD < 250 | 318.6–335.1 | 30 < SCD < 30 |
| transition w2/3 | - | - | interpolation | 75 < SCD < 171 |
| w3 | 360–390 | 250 < SCD | 323–335.1 | 171 < SCD |

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
