# Peer review of "Impact of SO2 Flux Estimation in the Modeling of the Plume of Mount Etna Christmas 2018 Eruption and Comparison against Multiple Satellite Sensors"

_remotesensing, doi:10.3390/rs15030758_

Round 1
Reviewer 1 Report
The paper of Lamotte et al. presents data about air pollution originating from the volcanic activity of Mt Etna volcano in 2018, acquired with very different techniques. In the present form the paper is very confusing and in my opinion better suited for a journal whose main scope is the study of atmospheric pollution.
One of the main issues is the lack of correlation with the type and intensity of volcanic activity. Some maxima may be correlated with increased volcanic activity but no data are presented to sustain their hypothesis. It would be very important to present data on SO2 emission showing its variation throughout the entire eruptive period. Another important parameter would be the variation in time of the height of the eruptive column.
Furthermore, notwithstanding that the authors recognise the importance of the wind direction for the dispersion of the volcanic contaminants they only consider this factor in the regional atmospheric circulation and not in the effects on the local air quality.
Several values are given in the text; however, what is missing is comparison with the climatological values. For example we have to know over specific locations the SO2 enhancements by means of percentage. Only then we will be able to say how much affected the volcanic eruption the local metrics.
I believe there should be a comparison with results from other studies. Was the height of the volcanic plume exceptionally large? In general, the discussion of this study's results should be done in accordance to other studies using similar tools.
Ref:
Babu, S. R., Nguyen, L. S. P., Sheu, G. R., Griffith, S. M., Pani, S. K., Huang, H. Y., & Lin, N. H. (2022). Long-range transport of La Soufrière volcanic plume to the western North Pacific: Influence on atmospheric mercury and aerosol properties. Atmospheric Environment, 268, 118806.
Filonchyk, M., Peterson, M. P., Gusev, A., Hu, F., Yan, H., & Zhou, L. (2022). Measuring air pollution from the 2021 Canary Islands volcanic eruption. Science of The Total Environment, 849, 157827.
Wang, X., Boselli, A., D’Avino, L., Pisani, G., Spinelli, N., Amodeo, A., ... & Stohl, A. (2008). Volcanic dust characterization by EARLINET during Etna's eruptions in 2001–2002. Atmospheric environment, 42(5), 893-905.
Arnalds, O., Thorarinsdottir, E. F., Thorsson, J., Waldhauserova, P. D., & Agustsdottir, A. M. (2013). An extreme wind erosion event of the fresh Eyjafjallajökull 2010 volcanic ash. Scientific reports, 3(1), 1-7.
Reviewer 2 Report
This manuscript has presented an interesting study on the measurements and modelling of volcano SO2 emissions by Mount Etna in 2018. The paper is detailed and well-organized. I think this manuscript is qualified to be published.
Generally, consistent spatial distributions of SO2 plumes were found between multiple simulations and observations, despite larger discrepancies in values. It seems that the model simulations using SEVIRI-derived emission fluxes have closer values with TROPOMI. Perhaps it would be good to discuss more on both methods, and their uncertainties sources of the derivations of SO2 emission flux from FLAME and SEVIRI. Also, more investigations and explanations on the discrepancies of emission flux are well welcomed.
Reviewer 3 Report
The paper "Impact of SO2 flux estimation in the modeling of the plume of Mt Etna Christmas 2018 eruption and comparison against multiple satellite sensors" by Lamotte C. et al. shows SO2 flux simulations of the Mount Etna eruption occurred in the last days of December 2018 performed with the MOCAGE model. The paper compares the output of the model with overpassing satellite measurements. In my opinion, it deserves to be published only after some specific points are addressed:
P6Figure1: what is the cause of the large disagreement between FL and SV e.g. for the 27th of December? Also, the FL data have many missing periods, what is the advantage of using these data if they are so fragmented? Would be possible to combine the two datasets (e.g. using FL when available and SV when FL are not available)?
P8Figure3: it may be worth merging the IASI_A and IASI_B data in order to obtain only one map at 8:00 UTC.
P9L347: What is the difference between TROPOMI_MPIC and TROPOMI_OP data? which difference should we aspect in the results and what the _MPIC data may add with respect to the operational data?
P15L508: A description of the TROPOMI_MPIC algorithm should be included in the paper, highlighting the differences w.r.t. the TROPOMI_OP product. The authors should also clearly motivate the reason for including the additional TROPOMI product TROPOMI_MPIC
P15L530: If it is possible to run the model with an increased horizontal and/or vertical resolution, why the authors did not use such configuration for this work? There are some limitations of the model which prevent that or it is just a limitation of the used computer facility?
P16L580: The authors are right in saying that the application of the averaging kernels (AKs) should lead to a more quantitative analysis. Indeed, I suggest to perform and add in the paper this analysis using only the TROPOMI_OP, if in the other cases the AKs are not available.
Minor checks:
P2L75: a further point is needed to conclude the sentence, after a.s.l.
P5L123: there is a space a not need space between the word "mode" and ":"
P5L123: there is a space a not need space between the word "stations" and ":"
P4L161: instead of "because...", in my opinion, is clearer to say: "after the first measurement period, during the year 2019 its spatial resolution has been..."
P14L484L acronyms FL and SV have been described before in the text.
P15L538: Please uniform "Etna", "Mount Etna" or "Mt Etna" throughout the paper.
Round 2
Reviewer 2 Report
the authors have addressed well my concerns
Reviewer 3 Report
The authors have addressed all my comments. The manuscript is ready for publication.